# Next-Generation Adjuvants: Applying Engineering Methods to Create and Evaluate Novel Immunological Responses

**DOI:** 10.3390/pharmaceutics15061687

**Published:** 2023-06-08

**Authors:** Jeremiah Y. Kim, Matthew G. Rosenberger, Nakisha S. Rutledge, Aaron P. Esser-Kahn

**Affiliations:** Pritzker School of Molecular Engineering, University of Chicago, 5640 South Ellis Avenue, Chicago, IL 60637, USA; jeremiahkim@uchicago.edu (J.Y.K.); mrosenberger@uchicago.edu (M.G.R.); nrutledge@uchicago.edu (N.S.R.)

**Keywords:** vaccine, adjuvant, adjuvant discovery, immunomodulator, systems vaccinology

## Abstract

Adjuvants are a critical component of vaccines. Adjuvants typically target receptors that activate innate immune signaling pathways. Historically, adjuvant development has been laborious and slow, but has begun to accelerate over the past decade. Current adjuvant development consists of screening for an activating molecule, formulating lead molecules with an antigen, and testing this combination in an animal model. There are very few adjuvants approved for use in vaccines, however, as new candidates often fail due to poor clinical efficacy, intolerable side effects, or formulation limitations. Here, we consider new approaches using tools from engineering to improve next-generation adjuvant discovery and development. These approaches will create new immunological outcomes that will be evaluated with novel diagnostic tools. Potential improved immunological outcomes include reduced vaccine reactogenicity, tunable adaptive responses, and enhanced adjuvant delivery. Evaluations of these outcomes can leverage computational approaches to interpret “big data” obtained from experimentation. Applying engineering concepts and solutions will provide alternative perspectives, further accelerating the field of adjuvant discovery.

## 1. Introduction to Adjuvants

Vaccines are considered one of the world’s greatest public health innovations [1,2]. The first vaccines used whole pathogens, which contain many endogenous molecules that activate the immune system. Modern vaccines are less frequently whole pathogens, but instead merely contain or encode for antigens, important pathogenic components. These modern subunit and nucleoside-modified mRNA vaccines no longer contain as many endogenous activating components. Instead, they often employ adjuvants: molecules added to the vaccine that stimulate the immune system [3]. Adjuvants work though a complex interplay of innate sensors. Generally, adjuvants use pathogen-associated molecular patterns (PAMPs) or damage-associated molecular patterns (DAMPs) to signal through important innate immune pathways such as the nuclear factor kappa-light-chain-enhancer of activated B cells (NF-κB) pathway or the interferon regulatory factors (IRF) pathway. The first adjuvant, alum, has been used clinically since the 1920s with great success [4]. Only in the past 20 years, however, has adjuvant discovery expanded substantially beyond alum, using new tools and knowledge about the immune system. To date, fewer than 10 adjuvants have been used in commercial vaccines [3,5,6]. Our goal is to provide some perspective on the future discovery of next-generation adjuvants. These are ideas and viewpoints, and if we are overzealous at points, it is due to our passion on the topic and not overconfidence in our own authority. We wrote this intending to provide fresh perspective and with the hope of engaging in conversation with others on the topic.

## 2. Traditional Adjuvant Development

Adjuvant discovery is a labor-intensive endeavor [7]. The traditional adjuvant discovery pipeline begins by identifying a target pattern recognition receptor (PRR), such as TLR7/8 [8,9]. High-throughput screening is used to assay thousands, sometimes millions, of molecules for their potential to agonize the target receptor. After many rounds of iterative screening, top “hits” of activity are identified as lead compounds. These lead compounds may be modified as per structure activity relationships for optimization. Final lead compounds are formulated with an antigen in a vaccine before rigorous testing. Rightfully so, adjuvants and vaccines have stringent safety requirements [3]. Thus, significant murine and non-human primate in vivo experiments are necessary before human clinical trials can begin. The challenge is that despite many successful adjuvant development methods, it remains difficult to elicit a desired immunological phenotype while predicting reactogenicity and tolerability in humans [10].

## 3. Engineering New Outcomes

We propose the next generation of adjuvants will require a new discovery pipeline, focused on screening for the end outcome: immunological activity. With many immune pathways and agonists identified, adjuvant development can now focus on identifying a phenotypic response to an adjuvant. Here, we discuss which aspects of adjuvanticity may be most important to engineer.

### 3.1. Target Identification and Signal Engineering

Adjuvants will benefit from selectively modifying signaling pathways to elicit desired immune outcomes. Instead of designing more potent receptor agonists, we believe more diverse results may be generated by “drugging” other aspects of the innate immune response. Apart from PRRs, other cellular signaling pathways, such as metabolism and proliferation, have been linked to immune outcomes. Innate pathway activation results in fate-specifying cytokines which control helper T-cell polarization. Adjuvants impact this elicited cytokine milieu, and tuning this process gives researchers greater control over adaptive responses. Cytokines are also, in part, responsible for vaccine reactogenicity. Optimizing for the reduction of cytokines associated with excess inflammation could alleviate adverse side effects, improving the likelihood of clinical translation. We believe modifying traditionally targeted pathways, and exploring new pathways, can address all these innate related signaling problems.

#### 3.1.1. Alternative Pathways for Adjuvanticity

Adjuvant targets should continue to expand beyond ligands for traditionally targeted Toll-like receptors (TLRs). In the past decade, research efforts have discovered new pattern recognition receptors, such as the cytosolic sensor STING. Cyclic dinucleotides, such as cGAMP, have been identified as STING ligands and are attractive adjuvant candidates [11,12]. Additionally, the NLRP3 inflammasome responds to broad disruptions in homeostasis, and serves to recognize a variety of PAMPs and DAMPs [13]. Besides PRRs, other signaling pathways impact innate immune activation and could be exploited for future adjuvant development (Figure 1a). Cell death pathways in the form of apoptosis, pyroptosis, and necroptosis may result in increased cross-presentation by activated neighboring cells [14,15]. Metabolic and epigenetic pathways are altered in trained immunity, a newly emerging concept where innate immune cells exhibit “memory” via heightened activation upon re-exposure to even heterologous pathogens [16]. These pathways represent a vast pool for novel adjuvants, as many activators of these pathways are currently being discovered.

#### 3.1.2. T_H_ Polarization: Going beyond T_H_1, T_H_2, and T_H_17

Adjuvants influence the adaptive immune response through their activation of innate immunity. Innate immune cells secrete fate-specifying cytokines which, traditionally, are considered the driver of polarizing helper CD4+ T cells towards a certain effector state (T_H_1, T_H_2, etc.) [17]. Adjuvants can control T-cell polarization by inducing specific cytokines, but currently approved adjuvants are still limited in this capability. Some adjuvants, such as CpG 1018 (a TLR9 agonist), are potent T_H_1 inducers, whereas others, such as alum, are strong T_H_2 inducers [18,19]. T_H_1 CD4^+^ cells promote cell-mediated immune responses by secreting IFN-γ, TNF-α, and IL-2 [20]. T_H_2 CD4+ cells promote antibody responses by secreting IL-4, IL-5, and IL-13. T_H_1 responses are typically desired for prophylactic vaccines, but ideally, vaccines and immunotherapies could be uniquely tailored to the application. For example, applications outside of infectious disease could benefit from alternative T_H_ polarization. T_H_17 cells, characterized by the production of IL-17, are implicated in protection against extracellular bacteria, and are critical in host responses to infection by *Klebsiella*, *Pseudomonas*, *Salmonella*, and *Bordetella* [21]. T_H_17 cells may increase protection against tuberculosis via neutrophil recruitment and inflammation mediation [22]. The extent of T_H_17 polarization is important, however, as excess T_H_17 responses can also worsen TB pathologies [23]. Thus, fine control over T_H_ polarization is needed, not simply general shifts from one bias towards another bias. Fine control over T_H_ polarization is challenging. Current adjuvant development is focused on the initiation of signaling events through receptor–ligand interactions, with downstream effects such as T_H_ polarization left as an afterthought. Next-generation adjuvant discovery could change this paradigm, focusing on end immunophenotypes such as T_H_ polarization.

With expanded understanding of the helper T-cell compartment, the conventional nomenclature for T_H_ polarization may be limiting [24]. T_H_ polarization traditionally relies on the profile of secreted cytokines from helper T cells. What started as a two-part T_H_1/T_H_2 model expanded with the discovery of T_H_17 cells in the mid-2000s [25,26]. Additionally, CD4+ T cells that serve in unique roles, such as regulatory T-cells (T_regs_) and follicular helper T-cells (T_FH_), further complicate T_H_ classification [27,28]. Finally, many other T_H_ subtypes have been proposed in recent years, including T_H_9, T_H_22, and T_H_GM [29,30,31]. The number of potentially expressed cytokines makes the current classification system impractical. Instead, we suggest, for adjuvant purposes, classifying T_H_ polarization by the five master transcription factors, T-bet, GATA-3, RORγT, Bcl-6, and FoxP3 (Figure 1b) [17]. Expression of these transcription factors was once thought to be mutually exclusive, but recent studies show that co-expression of transcription factors is possible [32,33]. We envision classifying T_H_ polarization as a location in a multidimensional space with levels of transcription factor as the axis variables. Such a classification could either apply to the total helper T-cell response as an “average” phenotype, or it could also apply to individual helper T-cells. New vaccine outcomes could be sought by applying adjuvants which polarize T_H_ responses across different locations in this multidimensional space.

#### 3.1.3. Reducing Reactogenicity of Adjuvants

Reactogenicity is a major challenge in the approval of many new adjuvants [6]. Adjuvants must provide enough stimulation to provide protection, while avoiding excess activation resulting in adverse side effects. The inability to control this balance limits the number of clinically approved adjuvants and leads to the withdrawal of certain vaccines. Adjuvant associated local and systemic inflammation leads to side effects such as injection site pain, fever, and headache [10]. Inflammation can be tracked by measuring the pro-inflammatory cytokines, IL-6 and TNF-α, for 1–24 h after injection [34]. These cytokines have been extensively studied and their role in inflammation is well explained in these reviews [35,36,37,38]. Severe reactogenicity damages not only patient health at the individual level, but also public vaccine perception at the population level. Minimizing vaccine reactogenicity while maintaining efficacy should be a priority for next-generation adjuvants.

As such, determining reactogenicity information early in the adjuvant discovery process is critical, to ensure that lead compounds are identified quickly and screened for toxicity or inflammation-related side effects. Reactogenicity can limit mRNA vaccine doses as some mRNA and lipid nanoparticles induce excess inflammation [39]. Currently, mRNA vaccine reactogenicity is addressed via nucleoside-modified mRNA. It remains unknown, however, how lipid nanoparticles activate innate immunity. Many hypotheses have been suggested, from direct sensing of particles to indirect sensing of particle-induced cellular damage [40]. Identifying such innate immune mechanisms is critical to identify effective adjuvants with reduced reactogenicity. With knowledge of the appropriate signaling targets, adjuvants could be screened for reduction of pro-inflammatory cytokines in combination with conventional lipid nanoparticle mRNA vaccine systems for applications such as COVID-19 and Lyme disease [41,42]. Other recently approved vaccines include saponins as adjuvants [43,44]. These plant-derived small molecules provide potent immunostimulatory effects, but also cause a high degree of toxicity [45]. This limits the effective dose of these adjuvants. Additionally, sourcing saponins is extremely difficult, though advances in its biosynthesis have been made [46]. Even with newer vaccine technologies, the need for potent, yet tolerable adjuvants remains.

Our group has been working to decouple inflammation from adjuvanticity using immunomodulators [47,48]. These immunomodulators are molecules which, in combination with known PRR agonists in a vaccine, decrease systemic proinflammatory cytokines and increase antigen-specific antibodies (Figure 1c). This effect applies across multiple PRR agonists and antigens, as the immunomodulators target intermediate signaling molecules downstream of receptor binding. Immunomodulators and other molecules which alter pattern-recognition-receptor signaling might be used to reduce reactogenicity in next-generation vaccines.

### 3.2. Formulation and Targeted Delivery

Next-generation adjuvants are often formulated to enhance delivery and efficacy. Early approved adjuvants did not target cells of interest, mostly using suspensions and heterogenous formulations. However, recently approved adjuvants often cannot rely on these methods. Adjuvant systems, such as GlaxoSmithKline’s AS series, contain many components formulated into nanostructures [49]. Another example is NovaVax’s Matrix-M adjuvant, in which immunostimulatory saponins are formulated with phospholipids and cholesterol [50,51]. As adjuvants become more complex, we argue that formulation should be considered earlier in development and discovery. Expanding the diversity of adjuvants necessitates trafficking to a variety of subcellular locations to aid receptor binding. Target receptors may be on the outer membrane, endosomes, or cytosol. Formulations can traffic adjuvants to these locations. For example, cationic lipid nanoparticles aid endosomal escape, greatly enhancing STING adjuvant activity, as this innate receptor is located in the cytosol [52]. Nanoparticle size is an important and simple factor to control. For example, 10–100 nm particles are the most efficient in lymphatic drainage, allowing facile diffusion to key secondary lymph organs [53,54]. Inside of this size range, particles of 50 nm are key in inflammasome activation as they aid in endosomal escape [55]. Alternatively, larger, micron-sized particles show reduced cellular uptake, limiting efficacy [56].

Formulations can also include external targeting ligands to aid delivery to the desired cell types. While adjuvant receptors are expressed primarily by immune cell types, bystander cells are sometimes affected by adjuvants. For example, arterial smooth muscle cells express TLR4. Activation of these cells promotes proinflammatory phenotypes that are linked to atherosclerotic lesions [57]. Thus, off-target effects on bystander cells should be limited. Not only do targeting ligands decrease side effects, but they also increase efficacy. This strategy is used to target obvious cell types, such as dendritic cells with DEC-205 and DC-SIGN, but we postulate that more niche cell types could benefit from targeted delivery [58,59,60]. Lymphatic endothelial cells, stromal cells that are implicated in long-term antigen storage, could be targeted via the VEGFR3 receptor [61,62]. First responder cells, dendritic cells that excel at phagocytosis of microstructures, could be targeted via the DAP12 and PRG2 receptors [63,64]. With increasing knowledge about innate immunity, we anticipate that formulation will play an increasing role in directing adjuvant responses.

## 4. Engineering New Evaluation Methods

With a greater number of approaches to discovery, an improved diagnostic framework is needed to evaluate new adjuvants. As the adage goes, “You get what you screen for”. Over time, traditional screens have not changed much, relying on laborious in vitro assays and synthetic modifications before preclinical studies in mice and non-human primates. This process of adjuvant discovery has been cited as one of the slowest processes in medicine [7]. If we want to achieve new outcomes, we must consider changing “what we screen for”. Currently, translational results take years to obtain. Here, we showcase alternative frameworks to evaluate adjuvants for early correlates of vaccine efficacy. These frameworks can potentially accelerate evaluation and reduce costs of adjuvant discovery. New computational approaches will aid in early stage screening, while rethought animal models will aid in preclinical trials.

### 4.1. Computational and “Big Data” Approaches

Technological and computational advances of the past few decades allow for restructuring of adjuvant discovery. Vaccine and adjuvant design is not limited to the simplistic “guess-and-check” empirical methods of the past. Instead, newer assays can provide massive amounts of data, assisting researchers with mechanistic understanding and efficacy evaluation [65]. Next-generation sequencing and other omics approaches allow for system level data collection. Machine learning and computational power enable in silico binding simulations [66]. High-content imaging and processing yield hidden patterns and features. Combined, these computational approaches can be implemented at all stages of adjuvant design—altering the workflow of adjuvant development.

#### 4.1.1. Systems Vaccinology

In the age of “big data”, we believe adjuvant discovery will benefit from a systems vaccinology approach that integrates large amounts of information. Compared to 20 years ago, immunological assays generate results with vastly improved speed and sensitivity. This allows researchers to collect data from numerous different experiments instead of relying on singular indications such as antibody titers. Systems vaccinology looks to consolidate information from high-throughput assays on cytokines, proteins, metabolites, gene expression, and other screens to obtain a comprehensive understanding on how an adjuvant affects a vaccine (Figure 2). A multidimensional approach consolidates useful data across in vitro and in vivo experiments to aggregate a snapshot of innate and humoral responses.

Recently, researchers have used a systems vaccinology approach with great success. For instance, the transcription factor CREB1 was shown to be a critical element for HIV-1 vaccines through a series of UMAP depictions of transcription-factor genes [67]. The Sampa lab used UMAP to leverage complex cytokine and transcriptomic datasets in a refined model of CREB-1 mediated HIV vaccine efficacy. At an organism level, the Chevrier lab tracks antiviral genes and tissue-resident memory T cells across most major organs after poxvirus vaccination [68]. Researchers can apply the mechanistic and toxicology insights from organ-specific readouts towards new adjuvants. At the population level, Dr. Ofer Levy applied a systems approach through a precision vaccines program. This program profiles vaccine-induced responses from different population groups, especially the young and the elderly, working towards effective personalized vaccines [65]. In the context of adjuvant development, incorporating data from children and elderly populations (either through PMBC donors or aged mice) into traditional high-throughput screens would allow for adjuvants that can affect specific population groups differently. Recently, this methodology was used to identify that a carbohydrate fatty acid monosulphate oil-in-water adjuvant enhances immunogenicity in aged mice [69]. A systems vaccinology approach has also been used to investigate predictors of antibody response across multiple vaccines [70]. As adjuvant and vaccine discovery continues to evolve, researchers should use the plethora of immunological data that we can obtain to increase our understanding of the whole response to a vaccine and adjuvant.

#### 4.1.2. In Silico and Machine Learning

Computational simulations of biological phenomenon, known as in silico experimentation, can improve the speed and accuracy of adjuvant discovery. Machine learning, an artificial intelligence methodology that builds algorithmic models to make probabilistic predictions, is often paired with in silico experimentation. Both of these techniques have found utility across chemistry, material science, and immunology [71,72,73]. For example, in silico protein modeling efforts identified small molecule antagonists that served as adjuvants to *Mycobacterium tuberculosis* and *Plasmodium yoelii* subunit vaccines in vivo [66]. Antagonizing CCR4 improved the immune response by reducing the inhibitory activity of regulatory T cells. This approach used molecular docking and homology modeling, traditional approaches that are commonplace in the drug discovery field [74]. More recently, researchers have implemented black-box optimization on in silico and in vitro screens. Black-box optimization is a machine learning technique wherein the models and constraints of a dataset are largely unknown. In drug discovery, the algorithms in this technique might take inputs such as small molecule structure and experimental results to output key predictive properties of successful drugs. Three common black-box algorithms include Bayesian optimization, reinforcement learning, and active learning, summarized in a review by Terayama et al. [75].

While finding expanded use in other fields, machine learning has yet to be applied at scale in adjuvant discovery (Figure 3). Our lab, however, recently applied Bayesian optimization, a Gaussian process regression that predicts expected properties, to high-throughput screening of adjuvants. For this screen, we used wet lab experiments to iteratively feed into a machine learning latent space. In our estimation, using this in vitro/in silico machine learning cycle reduced the experimental time and financial cost needed to perform the screen by tenfold (Figure 3A). This methodology of combining in silico libraries and machine learning algorithms allows for greater throughput of experimentation.

Machine learning in the adjuvant discovery space could also increase the number of successful compounds that pass major transition points in the discovery pipeline. Many in silico screens have failed to replicate computational results upon in vitro validation. Similarly, many compounds have failed to translate from murine models to non-human primates [76,77]. These are large challenges that machine learning may be able to help “de-risk”. Using multidimensional systems vaccine information and meta-analyses, machine learning could identify correlations between hit compounds that would not otherwise be apparent, yielding higher quality and number of compound hits. (Figure 3B). Additionally, machine learning algorithms could search already established datasets, such as those with drugs that did not pass clinical trials, to repurpose them for alternate uses such as adjuvants (Figure 3C). As the field of adjuvant discovery continues to develop, greater collaboration with machine learning groups can help provide new methods of understanding data and improving outcomes. In our limited experience, one can conceive of machine learning not as a replacement for current methods, but as an accelerant. As computers can be a “bicycle for the mind”, we suggest that adjuvant developers consider machine learning as a research assistant that can cut out the labor of screening and provide further clarity on projects where large-scale correlation is helpful.

#### 4.1.3. High-Content Imaging

High-content imaging allows for the visualization and quantification of therapeutic interactions in cell populations using multiparameter algorithms and fluorescent labeling [78]. High-content imaging has primarily been used in other drug discovery applications, but these techniques can be applied towards adjuvant discovery. Imaging has historically been limited by spatiotemporal restrictions and practicality. Recently, however, improvements in the technology now allows for improved visualization of three parameters: temporal dynamics, spatial organization, and intracellular interactions (Figure 4). By visualizing temporal dynamics, high-content imaging was used to develop an algorithm to screen for six parameters of hepatoxicity. This technique predicted high-risk drugs that potentially induce liver injury [79]. Similar algorithms could help screen adjuvants for adverse effects as the mechanisms of toxicity are not fully understood [10]. For monitoring spatial organization, researchers currently use multi-cellular models to recapitulate tissue repair, observe tumor growth, and quantify therapeutic interactions in 3D [80]. Reid et al. used high-content imaging to develop an algorithm to track changes in GFP expression levels for individual z-stack planes of a multicellular tumor spheroid in real time [80]. In the context of adjuvants, this will aid in studies using organoids to study immune phenomena such as germinal center reactions. We discuss organoids more in depth in the following section. Finally, highlighting the potential for monitoring intracellular effects, recent studies identified R788 disodium as an activator of the NF-κB pathway. The gradual entry of Rel-B into the nucleus after treatment with R788 was measured in a direct application of high-content imaging for adjuvant discovery. Meanwhile, this same technique was used to measure the inhibition of NF-κB via treatment with dehydrocostus lactone [81]. Studies using alternative delivery systems can visualize and measure cellular irregularity, antigen distribution, and cell activation [82,83,84,85]. As it relates to adjuvant discovery, high-content image analysis improves the optimization of screening compounds by predicting mechanisms of action, adverse effects, and improving accuracy. Using this technique, researchers can develop various assays to visualize the biodistribution and cytotoxicity of potential adjuvants in 3D multi-cellular models.

### 4.2. Rethinking Animal Models

Animal models have been the foundation of vaccine immunogenicity studies for centuries, yet there is a concerning lack of translation between preclinical animal work and clinical human results [87]. Mice are the most common animal model due to their cheap cost, easy husbandry, and human-like immune systems. However, while similar, the mouse immune system is not identical to humans. Mice express a variety of different PRRs compared to humans [88]. In mice, TLR8 responds differently than the human counterpart, though the two receptors are highly conserved [89]. The addition of human TLR8 into transgenic mice provides a better model for studying inflammation [90,91]. Similarly, mice lack the human TLR10, but transgenic mice expressing the receptor serve as a model to understand the negative regulation of TLR signaling [92]. Even among commonly held receptors, expression patterns on cell types varies between species. For example, TLR9 is found primarily on macrophages and myeloid dendritic cells in mice, but found primarily on B cells and plasmacytoid dendritic cells in humans [93]. These are examples of differences of a singular class of receptors on innate immune cells.

Humanized mice are attractive, yet complex options to solve the difference between species [94]. NOD/SCID/IL2rγ^null^ (NSG) mice are the most common immunodeficient mice for humanization efforts [95]. These mice can be reconstituted with human hematopoietic stem cells to attempt to recreate the human immune system. Still, antigen-presenting cells produce widely different MHCs across species [96]. While efforts to make NSG-HLA-transgenic mice exist, it is infeasible to fully recreate all the intricacies of the human system [97]. Non-human primates provide an alternative model, but at a steep economic and ethical cost. Instead, actual human systems, such as organoids, provide better engineering solutions for clinical relevance.

When attempting to replicate human physiology, primary cells and organoids are perhaps the closest available choice. Recent studies have successfully performed high-throughput screens on primary human peripheral blood mononuclear cells (PBMCs) [98,99]. PBMCs are non-trivial to source, but recent miniaturization of assays allows for a larger number of experiments to be performed for the same number of cells [100]. Using primary cells, however, enhances assay variability and is particularly sensitive to effects of donors [98]. Donor effects can be mitigated through increased sample size. Primary cells can provide some cellular interactions during standard tissue culture, but these co-cultures do not provide three-dimensional structural elements. Proper cellular organization is important in lymphoid structures, especially in germinal centers. Germinal centers are microstructures formed during an adaptive immune response, resulting in the creation of high-affinity antibodies [101]. Adjuvants can heavily influence the germinal center reaction, especially in enhancing follicular dendritic cell antigen deposition [102]. As an alternative to in vivo experiments, human organoids allow for the study of organized tissues, such as germinal centers, ex vivo. Organoids are made of human cells that organize three-dimensionally and are a useful tool for mid-throughput experimentation [103]. While the use of organoids in immunoengineering has risen in the past years, the application of these systems towards adjuvant research has been minimally explored and should be further adopted [104,105,106].

## 5. Engineering New Applications

Adjuvants have other applications apart from prophylactic vaccines against infectious disease. Cancer vaccines often employ adjuvants, but the desired immune response is different from infectious disease vaccines. Cancer treatments require increased cellular-mediated immunity in the form of tumor-infiltrating CD8+ T cells, as opposed to humoral immunity in the form of neutralizing antibodies. Thus, cancer vaccines, especially therapeutic cancer vaccines, need potent adjuvants to induce an appropriate innate cell phenotype in suppressive environments. Early trials to modulate the suppressed signaling environment focused on GM-CSF production, while recent attempts have shifted to modulating a broader repertoire of cytokines [107]. For example, AS15, a liposomal adjuvant including QS-21, MPLA, and CpG7909, promoted better cellular responses against the melanoma tumor-specific antigen MAGE-A3. In the clinical trial, this improved immune response was likely attributed to the increase in IFN-γ and IL-12 secreting dendritic cells [108]. We envision that future adjuvants will be developed with more control over innate cell activation to better match unique disease states.

Adjuvants do not have to be restricted to PRR agonists, but instead should include all molecules that “help” the immune system. In cancer, adjuvants might directly inhibit cancer cells, restoring immune function. Due to the vital role of cell death, proliferation, and growth in cancer, we believe researchers should screen for adjuvants that affect cell cycle checkpoints. Adjuvants such as cisplatin target tumor suppressor proteins that regulate cell cycle proteins, such as cyclins and cyclin-dependent kinases, thus causing cell cycle transition delays. Cyclin-dependent kinases (CDK) normally exist in an inactive state, but once activated, they form complexes with cyclins to signal cells to transition from one cell cycle phase to the next. Adjuvants that target cell cycles have been previously used in clinical trials for breast cancer. The use of these adjuvants paired with standard endocrine therapies showed substantially increased survival in patients.

Additionally, adjuvant drug discovery showed therapeutic potential when targeting cell senescence [109]. Senescence is when cells no longer proliferate due to DNA damage, remaining in the G_0_ resting state. Current tumor therapies induce the generation of senescent cells. Senolytics and senostatics, drugs that kill senescent cells or inhibit senescent paracrine signaling to other cells, have been identified as a potential second-line adjuvant therapy for current cancer treatments that cause DNA damage [110,111,112]. Recent high-throughput screens identified several senolytic compounds. ABT263, a novel senolytic drug, is anti-apoptotic and kills senescent cancer cells regardless of how senescence was caused. This compound was shown to inhibit cancer recurrence and substantially decrease metastasis [113,114]. Guccini et al. eliminated senescent cancer cells using senolytic BCL-2 inhibitors to prevent tumor cell metastasis [115]. Although the specificity and sensitivity of senolytics and senostatics still need further studying, we believe screening senolytics and senostatics as potential adjuvants allows for increased specificity for targeting different cell types when pairing with first senescent inducing therapies. The use of senolytics as adjuvants could potentially improve long-term survival of cancer patients.

Evaluating cancer vaccine adjuvants is complicated by their extensive use in combination therapies [116,117]. The most common combination immunotherapies employ checkpoint inhibitors, initially approved in 2011, to increase T cell activity. After preclinical animal studies, novel treatments are often added to FDA-approved treatments to create the combination therapy. While combination therapies often outperform monotherapies, it is hard to determine which treatments are responsible for the outcomes. Further complicating matters, cancer vaccines often fail to translate from preclinical mouse data to human clinical trials [118,119]. As discussed in the previous section, improving the evaluation of cancer vaccines and immunotherapies is critical. Mathematical methods have been suggested to model cytokine production in vaccine and checkpoint inhibitor combination therapies. Further development of these tools will aid in adjuvant development for cancer vaccines and other applications.

## 6. Conclusions

After decades of continuous progress, adjuvant research is expanding in many new, exciting directions. So far, adjuvant discovery has led to the approval of fewer than 10 adjuvants used in vaccine products. In the coming years, we believe engineering approaches will be integral to the discovery and development of a library of new adjuvants. By incorporating iterative design, phenotypic-driven high-throughput screens, and expertise from a wider range of scientists and engineers, we will obtain adjuvants that can more potently and specifically influence innate and adaptive immunity. As fundamental knowledge of innate immunology increases, we gain insight into how best to approach and design vaccines. With this knowledge, we can target novel pathways and modify existing signaling events. These approaches give researchers control over innate immune activation, allowing new adjuvants to develop improved outcomes, such as unique T-cell polarization and decreased vaccine reactogenicity. New adjuvants can be tailored to be used in specific population groups or for pandemic situations. Evaluating these outcomes of adjuvanticity, however, is no trivial task. Once again, engineering strategies provide improved diagnostics to aid in evaluation. With information-dense assays, computational tools including machine learning enable a systems-level approach to vaccinology. Combined with improvements in animal models, these approaches aid clinical translation of promising adjuvants by reducing time and costs of high-throughput screens, increasing the efficiency and quality of lead compounds, and understanding the mechanistic drivers of new adjuvants. With a greater ability to control the immune system, new adjuvants will be useful in nonprophylactic contexts. Future adjuvants can find new applications in combating immunosuppressive cancer environments and resistant infectious diseases. The next generation of adjuvant design is full of potential.

## Figures and Tables

**Figure 1 pharmaceutics-15-01687-f001:**
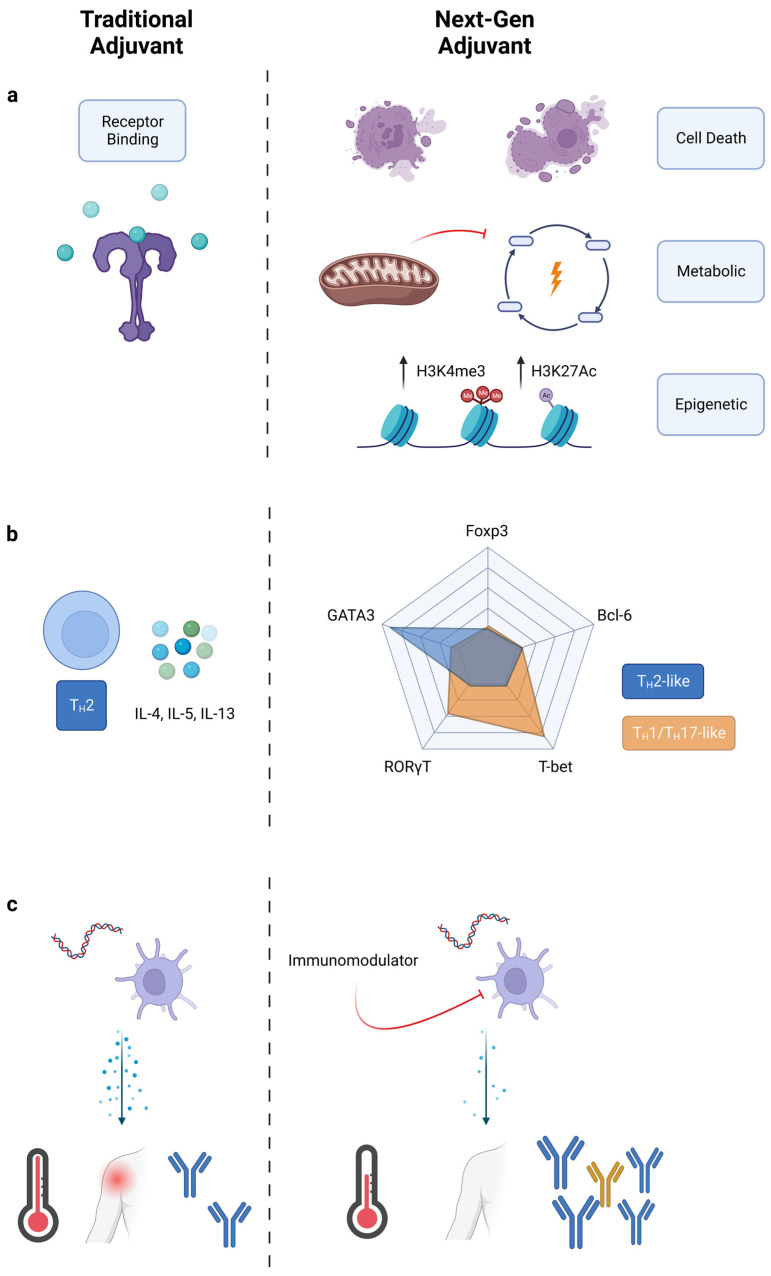
Engineering new immunological outcomes with adjuvants. Comparison of traditional adjuvants to next-generation adjuvants. (**a**) Next-generation adjuvants target pathways other than just PRRs. Cell death releases DAMPs, metabolic alteration improves cell sensitivity, and epigenetic modifications reprogram cells towards a trained immunity phenotype. (**b**) T-cell polarization shifted from cytokine classification to transcription factor classification. Next-gen adjuvants may create new immunophenotypes. (**c**) Traditional adjuvants elicit large amounts of inflammatory cytokines, while next-gen adjuvants reduce reactogenicity and improve adaptive responses.

**Figure 2 pharmaceutics-15-01687-f002:**
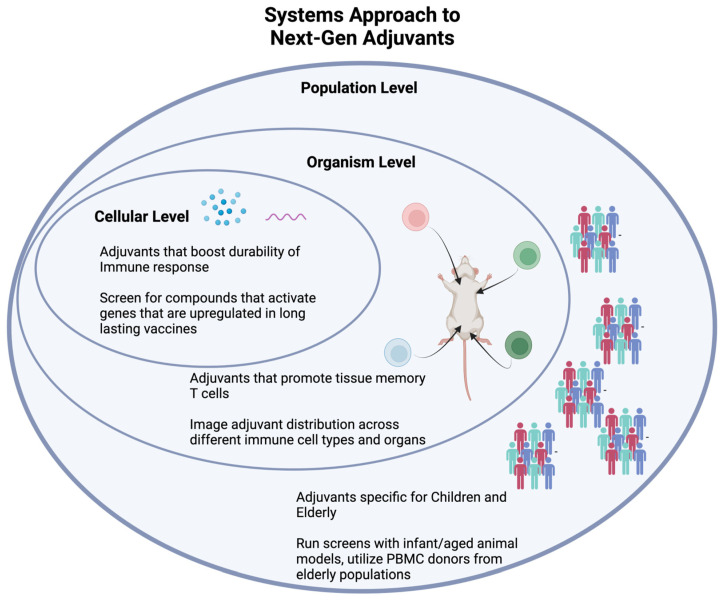
A systems vaccine approach uses new technology to understand the effects of an adjuvant at the cellular, organism, and population levels.

**Figure 3 pharmaceutics-15-01687-f003:**
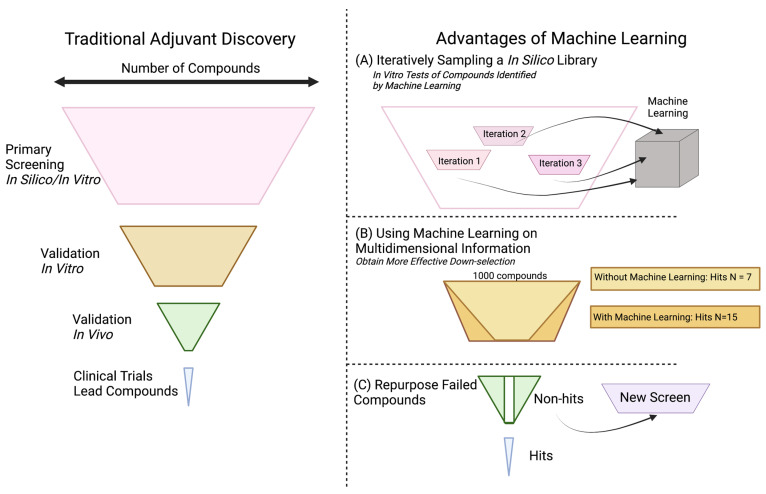
Machine learning and big data can (**A**) reduce costs of large screens, (**B**) identify common phenotypes for successful compounds, and (**C**) salvage late-stage compounds for alternative applications.

**Figure 4 pharmaceutics-15-01687-f004:**
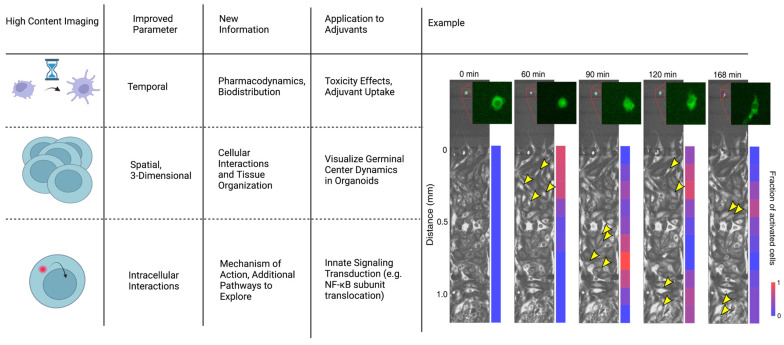
High-content imaging allows for more precise spatiotemporal understanding of how the adjuvant affects the immune system. Example incorporationg temportal, spatial, and intracellular interactions to understand NF-κB dynamics: Macrophage cytokine secretion dictates the signaling range and duration. Time lapse fluorescence images show coculture experiments, which imitate tissue infection. A single macrophage harboring p65-GFP was stimulated with LPS (10 ng/mL; green cell in the inset), which released TNF to activate NF-κB in the receiving population of cells (bottom chamber). Yellow arrow heads indicate examples of activated cells in each microscopy image, indicated by translocation of p65-GFP into the nucleus. The color bar on the right shows the activation rate of the receiving cells at increasing distances from the singular stimulated macrophage. Adapted with permission from Ref. [86]. 2022, Son, M. et al.

## Data Availability

Not applicable.

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
