# Peer review of "Next-Generation Adjuvants: Applying Engineering Methods to Create and Evaluate Novel Immunological Responses"

_pharmaceutics, 2023, doi:10.3390/pharmaceutics15061687_

Round 1

Reviewer 1 Report

Jeremiah and colleagues described a popular topic about adjuvants based on the current knowledge. They generally described the history of adjuvant use and the recent development. The manuscript focused on the new ideas of adjuvant development, such as new targets, new signaling pathways, new formulation and delivery methods, new evaluation methods and new screen methods. And, discussed some questions about animal models and new applications. These are what people interested in. This manuscript is well suited for publication in this journal with minor modifications.

1. The title of paper emphasizes on “immunoengineering.” However, to me, when talk about engineering, I think about we will do something to the immune system. Most contents of this manuscript described how to screen the target or pathways, which to me is not immuno-engineering, it is more like methodology-engineering. Therefore, I suggest the author to rethink about phrasing.

2. When talk about next generation, we think about “big data.” So, the authors should put more energy on computational approach. Please introduce more examples about “Machine learning” (AI) and some of the available methodology.

3. Figure 4 can not display what author want to express very well, please modify.

4. There are two “2.” On line 45 and 58. There is no “3.1”, the numbering start with “3.1.1.” on line 77.

Reviewer 2 Report

1.Figure legend "next gen adjuvants reduce reactogenicity and improve adaptive responses." Authors should be careful with words like reactogenicity which are vague.

2. Fig 2 legends title and Figure are confusing. What is the message here ? Figure does not depict New-generation engineering tools/

3. Figure 3 provides information on Imaging methods and their example but do not provide how this is in context to adjuvant design.

4. Figure 3: What is increased Hit Rate and Non-Hits ?

5. Abstract : Over the last 20 years, adjuvant discovery has been called laborious and slow.

Research shows there is more research in this area in last 20 years. What is the source of your information.

6. "Current adjuvant development consists of screening for an activating molecule, 11 formulating with an antigen, and testing in an animal model." Article English is not legible and often does not make sense.

7. Whole article is written in vague and nonscientific terms "Here, we consider new approaches using tools from engineering to improve next generation adjuvant discovery and development" What is engineering doing here ?

8. Abstarct talks about "new animal models developed from other research fields" What are the new animal models ? I am asking sicne article does not discuss any such model

9. In many places information is wrong. "In the past decade, research efforts have revealed agonists 79 for newly discovered pattern recognition receptors, most notably the STING ligand, 80 cGAMP11,12."

Do authors think cGAMP is receptor?

The article is written in vague terms and is facts are wrong in most places. The issue is not just grammar here but the incorrect usage of terms.

Reviewer 3 Report

Major

1.     Before conclusion, there should be a Future perspectives section where authors will provide new insights for readers and researchers.

2.     Conclusion section should be ended with some key  important facts and insights, rather than general discussion.

3.     In Figure 2, authors have mentioned “vaccines for youth and elderly”. Why authors have skipped children from it as they are one who vaccinated the most? Or authors suggesting the next-generation adjuvants integrated vaccines are for youth and elderly.

Minor

1.     In Introduction section, “Modern vaccines less frequently whole pathogens” should be “Modern vaccines less frequently used whole pathogens.

2.     pathogen associated molecular patterns” should be “pathogen-associated molecular patterns

3.     “damage associated molecular patterns” should be “damage-associated molecular patterns”

4.     Correct “which have been review adjuvants

5.     In section 2, “modified in structure activity relationships for optimization” should be “modified as per structure activity relationships for optimization

6.     Line no. 48-57 in Section 2 devoid citation of any reliable reference.

7.     There is heading numbering issue. There are two sections with No. 2.

8.     First paragraph in section 4.1 doesn’t have proper citations.

9.     humanlike” should be “human-like

10.   Terminologies like in vitro, in vivo, and in silico should be italicized, even in Figures. For instance, Figure 3

Authors are suggested to carefully recheck the manuscript to eliminate grammatical mistakes.

Round 2

Reviewer 2 Report

1. In my draft Figure 3 and 4 formatting is messed up. Please check it is corrected for formatting.

2. Authors should discuss lipid nanoparticle in case of COVID-19 vaccines and other mRNA vaccines (PMID: 34788080, 34862075).

3.  Does adjuvant size plays a role in immune reactivity ?

4. A table is needed in which major adjuvant activity is compared 

5.  Discuss adjuvant property of Saponin which is the key molecule in Matrix-M adjuvant

https://www.science.org/doi/10.1126/science.adf3727

Moderate editing of English language required

Author Response

Please see attachment, thank you so much!
